# A Guide for Oncologic Patient Management during Covid-19 Pandemic: The Initial Experience of an Italian Oncologic Hub with Exemplificative Focus on Uro-Oncologic Patients

**DOI:** 10.3390/cancers12061513

**Published:** 2020-06-10

**Authors:** Francesco A. Mistretta, Stefano Luzzago, Luigi Orlando Molendini, Matteo Ferro, Enza Dossena, Fabrizio Mastrilli, Gennaro Musi, Ottavio de Cobelli

**Affiliations:** 1Department of Urology, European Institute of Oncology, IRCCS, 20100 Milan, Italy; matteo.ferro@ieo.it (M.F.); gennaro.musi@ieo.it (G.M.); 2Risk Officer, Department of Medical Direction, European Institute of Oncology, IRCCS, 20100 Milan, Italy; luigi.molendini@ieo.it; 3Nurse Director, Multidisciplinary Surgical Area, European Institute of Oncology, IRCCS, 20100 Milan, Italy; enza.dossena@ieo.it; 4Chief Medical Officer, Department of Medical Direction, European Institute of Oncology, IRCCS, 20100 Milan, Italy; fabrizio.mastrilli@ieo.it; 5Department of Oncology and Haematology-Oncology, Università degli studi di Milano, 20100 Milan, Italy

**Keywords:** SARS-CoV-2, COVID-19, oncology service, medical oncology, surgical oncology

## Abstract

The recent exponential increase in the number of COVID-19 patients in Italy led to the adoption of specific extraordinary measures, such as the need to convey treatment of all non-deferrable cancer patients to specialized centres (hubs). We reported a comprehensive summary of guidelines to create and run an oncologic hub during the COVID-19 pandemic. Oncologic hubs must fulfil some specific requirements such as a high experience in oncologic patient treatment, strict strategies applied to remain a “COVID-19-free” centre, and the creation of a dedicated multidisciplinary “hub team”. Cancer treatment of patients who belong to external centres, namely spoke centres, could be organized in different pathways according to the grade of involvement and/or availability of the medical team of the spoke centre. Moreover, dedicated areas should be created for the management and treatment of patients who developed COVID-19 symptoms after hospitalization (i.e., dedicated wards, operation rooms and intensive care beds). Lastly, hospital staff must be highly trained for both preventing COVID-19 contagion and treating patients who develop the infection. We provided a simplified, but complete and easily applicable guide. We believe that this guide could help those clinicians who have to treat oncologic patients during the COVID-19 pandemic.

## 1. Introduction

The World Health Organization (WHO) recognized the Severe Acute Respiratory Syndrome Coronavirus 2 (SARS-CoV-2) and its associated disease (COVID-19) [1] as a pandemic on 11 March 2020. The number of infected patients and the mortality rate are rapidly increasing worldwide [2]. Clinical practice has, therefore, dramatically changed due to the need to treat all of the non-deferrable oncologic and benign conditions, while simultaneously protecting the safety of healthcare professionals and patients. Moreover, the urgent need to dedicate economic, infrastructural, medical and paramedical resources to patients with COVID-19 has generated a delay in all other non-urgent activities [3,4].

The recent exponential increase in the number of COVID-19 patients in Italy, and particularly in Lombardy, which is the most affected region in Italy and one of the most worldwide, led to the adoption of specific extraordinary measures [5,6]. Specifically, the concept of “personalized medicine” has rapidly shifted to the one of “population medicine”, namely: treating all non-deferrable patients, while avoiding the diffusion of the SARS-CoV-2 infection [7]. In consequence, there is an urgent need to properly define all non-deferrable oncologic conditions that has to be rapidly treated [8]. Moreover, cancer treatments should be conveyed to specialized “COVID-19-free” oncologic centres, where a specific attention must be given to healthcare professionals, in order to minimize the risk of exposure.

To address these unmet needs, the Italian National Healthcare system and Regione Lombardia organized specific oncologic hospitals, namely oncologic hubs, that have to provide care to all non-deferrable cancer patients during the COVID-19 pandemic. The aim of the current manuscript is to provide a specific guide to create, and to organize, an oncologic hub hospital.

## 2. Patients’ Risk Characteristics

Although oncologic conditions usually require a priority treatment, relative to benign pathology, it could be hard to choose who to prioritize among oncologic disease patients. The COVID-19 pandemic obliged the health care system to assess a risk characteristics chart in order to stratify patients according to tumour biology, minimizing the risk of disease progression and allowing each patient to receive the best survival benefit. We thus identified three different categories, based on the risk of disease progression and severity:

Red category (high urgency/severity), in which treatment must be delivered within 4 weeks: patients with high-risk aggressive tumours, or patients with tumour-related complications, for which treatment cannot be deferred.

Yellow category (intermediate urgency/severity), in which treatment must be delivered within 8 weeks: patients with intermediate-risk tumours, or patients that can receive bridging procedures for high-risk aggressive tumours, or patients with tumour-related complications, for which treatment can be deferred or that does not require hospitalization.

Green category (low urgency/severity), in which treatment can be postponed after 8 weeks: patients with low-risk tumours, or patients that can receive bridging procedures for intermediate-risk tumours.

It must be noted that this classification has been established during this emergency period and thus might be perfectible. In particular, hospital waiting lists, patient’s age and performance status, as well as differences in risk characteristics among different cancer pathologies, were not taken into consideration in this analysis. In consequence, these indications have to be taken with a grain of salt, and dedicated multidisciplinary figures should evaluate the possibility of also treating patients who, at first sight, may fall into lower risk categories.

In our case, the urologic malignancies included in the red category were: testis cancer, locally advanced kidney tumours (cT2 or higher), urothelial cancers, intermediate or high-risk prostate cancer (ISUP grade group ≥2), penile cancer that cannot be conservatively treated, tumour-related complications that cannot be deferred. Nevertheless, for example, it might be difficult to prioritize between a 6.5 (cT1b, yellow category) vs. 7 cm (cT2, red category) kidney cancer, or to discriminate between an elderly high-risk (red category) vs. a young intermediate-risk prostate cancer patient (yellow category).

## 3. Proposed Strategies to Decrease the Risk of Contagion

### 3.1. Ambulatory Patients

In order to decrease the risk of contagion for COVID-19 for ambulatory patients, it could be of use to adopt some preventive strategies. First, we perform a telephonic interview with all patients who had, in their program, access to our institution. The interview consists of three simple questions: (1) Did you receive a positive COVID-19 diagnosis? (2) Did you have fever and/or cough and/or dyspnoea during the last two weeks? (3) Did you have acquaintances with patients affected or suspicious for COVID-19? In case of at least one positive answer to these questions, hospital access must be postponed and general practitioner should evaluate the patient. 

Second, all patients who are supposed to access our institution are subjected to a preliminary triage at the entrance, where the same questions as the telephonic interview are repeated and temperature is tested. In case of fever (body temperature higher than 37.5 °C), hospital access must be postponed and a general practitioner should evaluate the patient. Furthermore, we provide surgical masks and gloves for all patients at the entrance, which have to be used throughout the entire duration of their hospital stay. 

Third, to reduce the exposition, relationship and traffic, it may be important to create itineraries and areas dedicated to specific tasks, such as hospitalization triage, outward medical visits and instrumental examinations.

Fourth, all ambulatory patients who are supposed to receive a follow-up examination or a diagnostic evaluation that could be deferred, should have the examination or evaluation postponed. Telephone consultation associated with remote imaging or laboratory analyses evaluation may be a valid deputy to medical examination in case of follow-up, and could heavily lessen the number of patients accessing the hospital [9,10].

Fifth, only one companion for each outpatient must be allowed and should be limited only to non-self-sufficient patients. Companions must undergo, at the hospital entrance, the same preliminary triage as the patients. Again, companions should wear a surgical mask and gloves for the entire duration of their hospital stay. In consequence, surgical masks and gloves should also be provided to companions. 

Sixth, access to hospital comfort services (canteen, coffee shops etc.) should be limited to hospital staff members and only for limited time periods. 

For what concerns our urologic practice, we dismissed the frontal medical examination for follow-up of most prostate cancer patients previously cured with active treatment (i.e., radical prostatectomy or radiotherapy) and implemented phone interviews to collect prostate-specific antigen (PSA) values or patient symptoms. Similarly, we deferred all outward cystoscopic examinations of low-risk bladder tumours with no evidence of disease recurrence during the last 12 months. Conversely, we continued all the intravesical Bacillus Calmette-Guérin immunotherapy administration for patients at high risk of tumour recurrence. Moreover, due to the non-negligible risk of immunodeficiency after systemic chemotherapy administration, the urologic multidisciplinary board decided to avoid neoadjuvant chemotherapy treatment for muscle-invasive bladder cancer patients. Prompt radical cystectomy was therefore recommended. 

### 3.2. In-Hospital Surgical Patients

Routinely, surgical patients undergo preoperative examination before hospitalization. However, during the pandemic era, it could be possible that a patient may incur a contagion in the period between preoperative examination and hospitalization. To decrease the risk of hospitalizing infectious patients and to exclude active infections, in addition to the aforementioned triage, white blood cells count, C-reactive protein and lactate dehydrogenase blood levels are investigated in all patients within five days before surgery. In addition, chest radiography is performed in all patients whose preoperative examinations have been conducted thirty or more days before surgery (Figure 1).

During hospitalization day, the patient undergoes a double check triage at the hospital entrance and in the surgical ward. In case of fever (body temperature higher than 37.5 °C) or fingertip oxygenation lower than 95%, the admission is cancelled and the patient is sent home with COVID-19 infection suspicion, and followed for 14 days. Conversely, if vital signs and laboratory exams are normal, but the patient presents mild symptoms or anamnestic suspect of infection, a novel set of blood tests and a low-dose computed tomography chest scan are taken. All patients with findings of interstitial pneumonia on chest radiography or chest computed tomography must undergo a nasopharyngeal COVID-19 test. These patients must undergo quarantine, with written indication to immediately contact their general practitioner in case of symptoms. The COVID-19 test results will be communicated as soon as available. Lastly, hospitalized patients must wear a surgical mask and gloves for the entire duration of their hospital stay.

Ideally, all hospitalized patients in an oncologic hub should be tested for COVID-19 (nasopharyngeal test). Nevertheless, the scarce availability of the test generated by its massive request at the beginning of pandemic in Italy, made this strategy difficult to be fulfilled. Moreover, it may be possible that the commercialization of serological tests may change the screening modality. In consequence, the performance of the COVID-19 nasopharyngeal test should be prioritized in some specific categories of surgical candidates at highest risk of contagion spreading: (1) Thoracic surgery patients, (2) larynx and/or trachea surgeries or patients with tracheostomy, (3) patients for whom intensive care unit admission is planned, (4) emergency admissions. From 8 March until 23 April, 22 patients, who were candidates for surgery, were diagnosed with SARS-CoV-2 infection after a nasopharyngeal swab, and their hospitalization was annulled. As recommended by the Italian Healthcare Ministry and Regione Lombardia, all subjects positive for SARS-CoV-2 infection have been reported to their general practitioners and to the COVID-19 unit of crisis. These patients underwent home quarantine for at least two weeks. In case of symptoms development, they were hospitalized to other COVID-19 specialized centres. Conversely, asymptomatic patients were retested at the end of quarantine with a nasopharyngeal swab and, in case of two consecutive negative results, they were finally declared eligible for hub hospitalization.

Visits to hospitalized patients should be reduced as much as possible. Visits must be scheduled when the request for medical or nursing activity is low and the in and out hospital flow is at its lowest rate. Usually, these two conditions meet at late afternoon or early evening. Moreover, patient’s visits must be limited to only one guest per day (one hour a day), and should be forbidden for the most frail patients, such as those in onco-hematologic divisions. Again, all visitors must wear a surgical mask and gloves for the entire duration of their hospital stay.

## 4. Oncologic Hub

### 4.1. Rationale and Hub Characteristics

Oncologic diseases usually require a priority treatment, relative to benign pathologies. However, the COVID-19 pandemic obliged several hospitals to allocate most of the intensive care beds to COVID-19 patients or to convert wards and operating rooms into new subintensive or intensive care units. In consequence, most of the surgical activity of pandemic areas was suspended or deferred. In order to treat non-deferrable patients, such as emergencies or oncologic conditions at high risk for disease progression, the Italian healthcare ministry and Regione Lombardia centralized the treatment of these patients in high-experienced hospitals (Hubs). 

Oncologic hub hospitals must respect specific requirements. First, the oncologic hub must be a referral centre, with high surgical volume and experience in order to offer the best curative chances, with the lowest risk for complications and the prolonging of hospitalization. Second, oncologic patients are at higher risk of infection, relative to general population, due to the immunosuppression state caused by the oncologic disease, which could be worsened by surgical or systemic treatments [11,12,13]. In consequence, a strict protocol to maintain “COVID-free” surgical and onco-hematologic wards must be applied. Third, oncologic hub hospitals should have enough infrastructural, medical and paramedical resources in order to take care of all non-deferrable cancer patients that need treatments during COVID-19 pandemic.

### 4.2. Patient Selection

It must be taken into account that not all patients belonging to external centres, namely spoke centres, are eligible for treatment. Considering that only patients belonging to the red or yellow categories should be treated, all patients should be confirmed as a high-risk by members of the oncologic hub. Although tumour characteristics such as biology and staging are generally the most important features for risk assessment, other information must be obtained, such as a complete medical history and time from diagnosis. Moreover, patient origin has to be investigated to assess the risk of COVID-19 contagion and in order to evaluate the availability of patient for transfers. In consequence, a multidisciplinary “hub team” must be dedicated to an ideally holistic evaluation of patients belonging to spoke centres and routinely weigh the risk assessment of each patient included in the waiting list. 

The main characters of this multidisciplinary team are: surgeon/clinician, anaesthesiologist, risk officer, and the chief medical officer. In particular, the risk officer and the chief medical officer guarantee the correctness of the entire patient selection from the point of view of regulatory requirements. Moreover, a paramedic staff must be assembled in order to organize, schedule and coordinate the several steps of patient management from preoperative examinations to discharge. If necessary, the “hub team” may require additional preoperative tests or a new patient examination before surgery. Finally, it is important to note that the “hub team” may deem the patient unsuitable for treatment at the hub centre at any time. 

### 4.3. Management of Surgical Patient Belonging to Spoke Centre

For the surgical and perioperative management of patients coming from spoke centres, three different pathways could be planned, according to the grade of involvement and/or availability of the medical team of the spoke centre (Figure 2): (A) Exclusive management of oncologic hub staff, (B) shared management between oncologic-sub and spoke centre staff, (C) exclusive management of spoke centre staff. 

A- and B-category patients will be hospitalized in a specific medical ward according to the type of the tumour (i.e., urology for uro-oncologic patients). Conversely, C-category patients’ hospitalization will be in mixed surgical wards. 

In order to treat all non-deferrable oncologic patients coming from spoke centres, original oncologic hub waiting lists should be modified. Moreover, it may be possible that dedicated extra operating rooms must be organized. 

Given that our institution is an oncologic tertiary referral centre, the treatment of patients coming from other centres was routine in the past. In consequence, the treatment of A-category patients did not change our daily practice. The increased rate of B-category patients instead represented a novelty. However, we ascertained that knowledge exchange and daily clinic sharing between different teams might generate opportunities for professional growth. Lastly, no patients belonging to the C-category were treated at our institution until the completion of the current manuscript.

From 8 March until 23 April, 1569 patients were surgically treated at our institution for an oncologic disease. Of them, 267 harboured a urologic-oncologic pathology. Overall, 148 patients belonged to a spoke centre. Of them, 48 harboured a urologic-oncologic pathology.

### 4.4. Management of COVID-19 Patients

During this pandemic era, it is fundamental that an oncologic hub hospital should remain a COVID-19 free hospital. However, despite the best preventive measures, it could happen that patients who are asymptomatic at the time of admission will develop signs and symptoms of infection during hospitalization. COVID-19 infection should be suspected in patients with a sudden onset of respiratory symptoms and/or fever that are not supported by other possible aetiologies. These patients must undergo a COVID-19 nasopharyngeal swab. Conversely, confirmed COVID-19 cases are those with a positive COVID-19 test, with or without respiratory symptoms and/or fever.

It is of utmost importance to build dedicated areas for both suspicious and confirmed COVID-19 patients, where only deputed multidisciplinary medical and paramedic personnel should have access. A dedicated ward should have specific characteristics in order to isolate patients and avoid the spread of viruses among patients and hospital employees. Separate areas within this dedicated ward should be created in order to: (1) allow the observation/assistance of suspicious patients awaiting the results of the COVID-19 test (observation area); (2) assist confirmed COVID-19 cases (hospitalization area). The COVID-19 ward might be located in a corner of the hospital, with doors that can limit entrance and exit. Ideally, the ward should be situated far from onco-hematologic divisions, where most frail patients are hospitalized. Patients can be discharged from this dedicated COVID-19 ward in case of: (1) negative COVID-19 test and no more clinical symptoms (observation area); (2) two consecutive negative COVID-19 tests on two consecutive days (hospitalization area). Overall, 15 patients suspected of infection were transferred to the COVID-19 ward. Of them, three patients were diagnosed with the COVID-19 infection after a nasopharyngeal swab.

Dedicated areas for intensive care assistance should be identified or built. These intensive care units should have analogous characteristics of isolation and access must be limited to dedicated multidisciplinary personnel. For both COVID-19 wards and intensive care units, adequate individual protection devices, such as masks, disposable gloves and clots, should be located at the entrance, dressing and undressing areas must be sited before the COVID-19 area entrance, while waste bins to safely dispose of these devices should be put at the exit. Medical and paramedic staff must be adequately trained for the safe dressing and undressing. It is worth noting that the transfer of any COVID-19-positive patient to other hospitals should be carried out as quickly as possible, according to the patient’s clinical conditions. Moreover, dedicated COVID-19 itineraries and areas must be created for radiologic and endoscopic procedures.

Lastly, it may be necessary to surgically treat hospitalized patients who developed COVID-19 during their hospital stay. Again, dedicated operating rooms, instrumentations and staff are mandatory. In particular, operating rooms should be located at one extremity of the surgical facility, should have independent access, a negative pressure environment and must be provided with a ventilator used for oxygenation of only COVID-19 patients. Ideally, expert medical and paramedic staff should be chosen to permit a multidisciplinary use of the COVID-19 operating room. As previously suggested [14,15], during endotracheal intubation, the airway should be secured using the method with the highest chance of first-time success to avoid repeated instrumentation of the airway, including using a video-laryngoscope. Moreover, laparoscopic and robot-assisted surgical procedures could be associated with a higher risk of COVID-19 contagion, relative to open surgery [16]. Specifically, this scenario applies to surgical procedures at the level of the respiratory or gastro-intestinal system. In our case, the uro-oncologic procedure associated with higher risk of COVID-19 contagion is minimally invasive radical cystectomy with laparoscopic intestinal handling.

## 5. Hospital Staff Management

During the pandemic era, medical and paramedic staff might be both caregiver and contagion spreader at the same time [17]. In consequence, preventive strategies should be applied specifically for hospital personnel (Figure 3). In particular, cutaneous temperature must be evaluated everyday for all employees before entering the hospital. Moreover, in case of suspicion or borderline values, tympanic temperature must be tested to confirm the cutaneous results.

According to Italian National Health System recommendations, health care workers or other people employed in the assistance of a suspected or confirmed case of COVID-19 should not be considered “infected”, if the assistance activity has been carried out with the use of correct individual protection devices.

Asymptomatic workers who assisted a probable or a confirmed case of COVID-19 without the use of appropriate individual protection devices must contact the occupational medicine service. The same strategy applies for workers who had close contact with a probable or a confirmed case in the context of their private life. In these two categories, the COVID-19 nasopharyngeal test is not immediately recommended. However, clinical conditions of the workers must be monitored day by day (cutaneous temperature must be evaluated twice a day and any respiratory symptom should be taken into consideration). Asymptomatic workers should continue their assistance activity with the use of appropriate individual protection devices. Conversely, workers who developed respiratory symptoms and/or fever (body temperature higher than 37.5 °C) must stop working and should undergo a nasopharyngeal COVID-19 test and chest radiography. Subsequently: (a) Workers with a positive COVID-19 test must undergo quarantine for at least 14 days after the recovery from symptoms. Two consecutive negative COVID-19 tests on two consecutive days are necessary before resuming their daily job. (b) Workers with a negative COVID-19 test must undergo quarantine until the end of symptoms. One negative COVID-19 test is recommended before resuming their daily job.

Nevertheless, it must be of utmost importance to do not underestimate the risk of false negative nasopharyngeal test results. In consequence, during either daily clinic or desk activities, all hospital staff must always use adequate individual protection devices, not just for their safety, but also to limit inadvertent contagion spreading. This said, due to limited availability of individual protection devices in a “COVID-19 free” oncologic hub, the use of these devices should be optimized. Specifically, all hospital workers must wear a surgical mask and gloves. Conversely, only healthcare workers exposed to higher risk of contagion (i.e., aerosol or droplets emission procedures, COVID-19 ward etc.) must adopt more strict individual protections (FFP2/FFP3 masks, two pairs of gloves, disposable cap, protective glasses or visor and disposable water-repellent gown). The main manoeuvres that are considered at higher risk of aerosol emission are: nose/mouth/tracheal intubation, endotracheal aspiration, tracheotomy, change of tracheostomy tubes, bronchoaspiration, gastroscopy, analysis of fresh operating samples in the pathological laboratories, preparation of slides during needle aspiration.

In addition to the use of individual protection devices, all other measures that limit the spread of respiratory infections are recommended (frequent hand washing, avoid groupings, safety distances, etc.). Lastly, all services performed by workers from external companies must be carried out wearing surgical masks and gloves.

Phone or web conferences should be implemented. This technology may help to give continuity to multidisciplinary boards, journal clubs and students or trainees’ lessons [18]. Similarly, in order to avoid aggregation, cafeteria services, as well as canteens, must permit a distance of at least two meters between customers. Lastly, a psychological interview service must be created for hospital employees [19]. This service must be available both by phone call and online. 

## 6. Materials and Methods

On 8 March 2020, the Italian National Healthcare System and Regione Lombardia communicated the need to organize hub hospitals in order to provide healthcare to all non-deferrable diseases during the COVID-19 pandemic (DGR.2906, DG Welfare 3553) [6,20]. Separate hub hospitals were organized for non-cancer (i.e., stroke, heart attack, traumata etc.) and cancer patients. The European Institute of Oncology (IEO), due to its expertise in the treatment of oncologic patients, was therefore selected as an oncologic hub. Specifically, the IEO became an oncologic hub for the treatment of all non-deferrable cancer patients coming from 32 different hospitals (spoke centres) of Lombardy. The following project did not involve the ethic committee since data from individual patients were not collected nor processed.

A dedicated “hub team” was created to establish specific intra-hospital guidelines for treatment and management of all patients coming from spoke centres. 

The “hub team” was composed of several figures: (1) chief medical officer, (2) risk officer, (3) clinician/surgeon, (4) anaesthesiologist, (5) nurses, (6) case managers, (7) secretaries, (8) other hospital figures.

Guidelines for the selection, management and treatment of cancer patients at oncologic hub hospitals during the COVID-19 pandemic are provided within the manuscript. Specific examples for management of uro-oncologic patients are also reported.

## 7. Conclusions

The recent pandemic explosion of SARS-CoV-2 and its associated disease change daily practice for oncologic patients. In consequence, there is a need to convey cancer treatment to specialized centres. However, specific recommendations to manage and treat this patient category are lacking. 

We reported a comprehensive summary of guidelines to create and run an oncologic hub during the COVID-19 pandemic. First, an oncologic hub must fulfil some specific requirements such as a high experience in oncologic patient treatment, strict strategies applied to remain a “COVID-19 free” centre, and the creation of a dedicated multidisciplinary “hub team”. Second, treatment of cancer patients who belong to spoke centres could be organized in different pathways, according to the grade of involvement and/or availability of the medical team of the spoke centre. Third, dedicated areas should be created for management and treatment of patients who develop COVID-19 symptoms after hospitalization (i.e., dedicated ward, operation rooms and intensive care beds). Fourth, hospital staff must be highly trained for both preventing COVID-19 contagion and treating patients who could develop the infection. 

Our strategies led us to surgically treat more than fifteen hundred patients in around 45 days, limiting the missed diagnosis of COVID-19 patients before hospitalization to only three cases (~0.2%). Moreover, a fast response to the request for an oncologic-hub creation led us to treat around 150 high-risk oncologic patients belonging to spoke centres, numbers that are rapidly increasing day by day.

The current manuscript aims to provide a simplified, but complete and easily applicable guide that was born after several working hours and which has seen the participation of different figures for its completion. We believe that this guide could be of help for those clinicians who have to deal with oncologic patients during the COVID-19 pandemic. Nevertheless, we also believe that this guide could be perfectible and the results of our effort could be definitively assessed at the end of the pandemic.

## Figures and Tables

**Figure 1 cancers-12-01513-f001:**
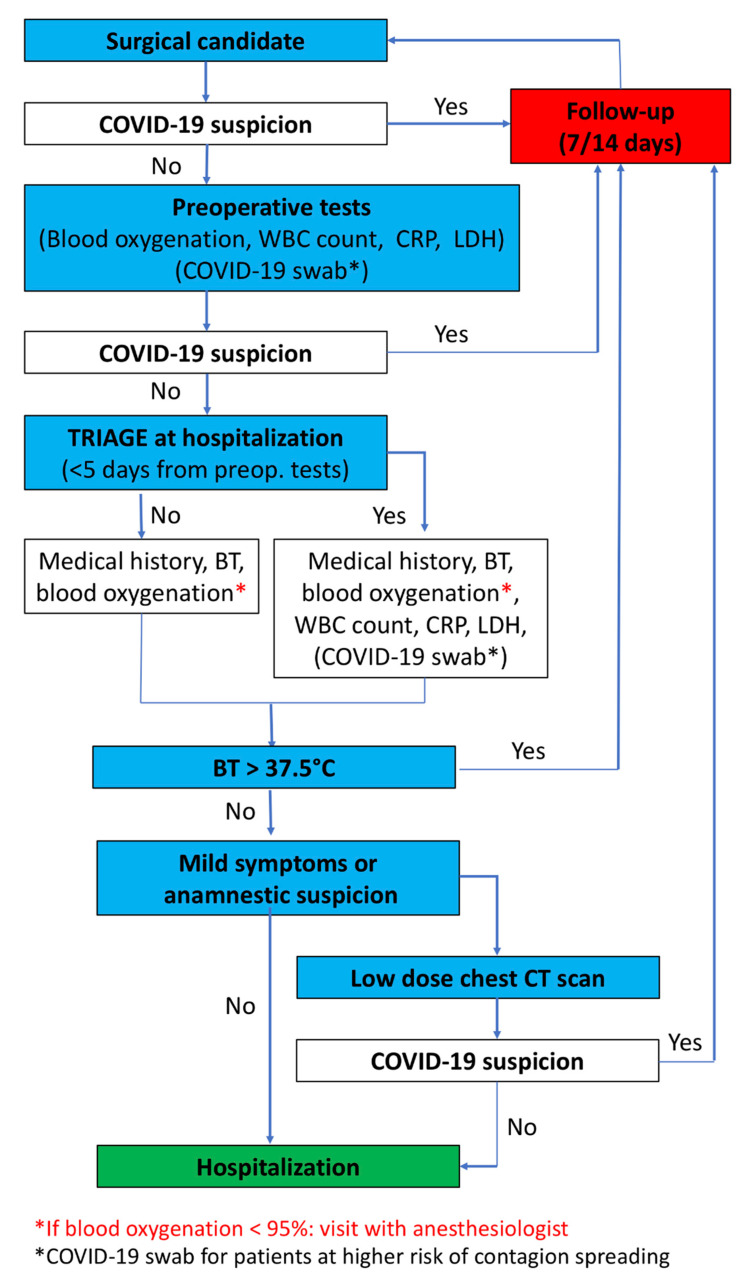
The flow chart describes proposed strategy of surgical patient management to decrease the risk for contagion during COVID-19 pandemic (abbreviation list: white blood cell (WBC), C-reactive protein (CRP), lactate dehydrogenase (LDH), body temperature (BT), computed tomography (CT)).

**Figure 2 cancers-12-01513-f002:**
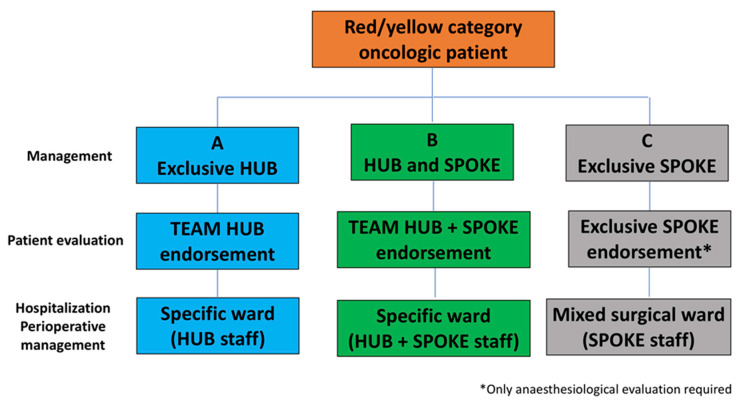
The flow chart describes the oncologic hub patient management stratified according to the grade of involvement and/or availability of medical team of the spoke centre.

**Figure 3 cancers-12-01513-f003:**
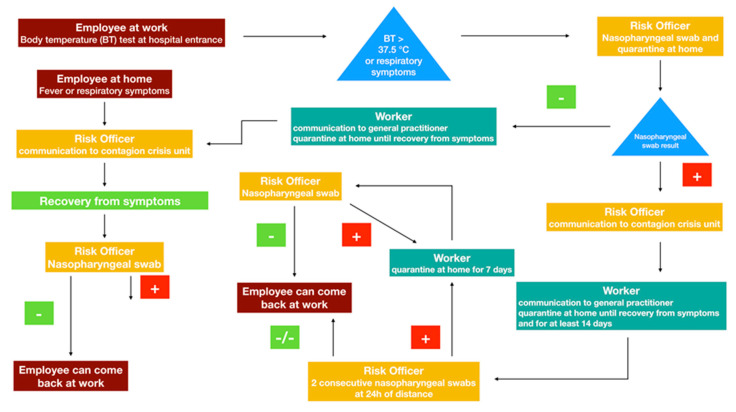
The flow chart describes the oncologic hub staff management during COVID-19 pandemic.

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
