# Peer review of "A Guide for Oncologic Patient Management during Covid-19 Pandemic: The Initial Experience of an Italian Oncologic Hub with Exemplificative Focus on Uro-Oncologic Patients"

_cancers, 2020, doi:10.3390/cancers12061513_

Round 1
Reviewer 1 Report
I have already reviewed this paper. The authors have appropriately responded to comments from three reviewers (revised version).
Reviewer 2 Report
I am glad that all my comments were addressed in the revised manuscript.
Reviewer 3 Report
It is great for publication after revision. I have no more comments.
This manuscript is a resubmission of an earlier submission. The following is a list of the peer review reports and author responses from that submission.
Round 1
Reviewer 1 Report
This manuscript describes the Italian experience of management of urologic cancer patients during Covid-19 pandemic.
A few comments;
- The authors mention "HUB" without any words which stand for it for several lines until line number 170. It would be nice if the authors mention what HUB stands for in the first page.
- The authors mention "SPOKE centers" several times. We, the readers from outside Italy, do not understand what SPOKE stands for. Please explain.
- I noticed the use of "outward patients" several times in the manuscript . I believe the authors refer to "outpatients" or "ambulatory patients". The usual English word is "ambulatory patients".
- Scientific soundness and interest to the readers is considered "low".
Reviewer 2 Report
The commentary article “ A guide for oncologic patient management during COVID-19 pandemic: the initial experience of an Italian oncologic HUB with exemplificative focus on uro-oncologic patients” provides a comprehensive detailed guideline for the treatment of oncologic patients during COVID-19 pandemic to minimize the risk of spread in hospitals.
Major comment:
Line “278-280”: all healthcare professionals who take care of suspected or confirmed COVID-19 patients should be required to use appropriate personal protective equipment (PPE). If those do not wear appropriate PPE during the care of suspected or confirmed COVID-19 patients, there is a higher chance that they will contract the viruses. As we now know that asymptomatic or pre-symptomatic COVID-19 cases can shred the infectious viruses in the upper respiratory tract, it is important to test these workers for COVID-19 especially if they also assist other non-COVID-19 patients.
Minor comment:
- Line 17, please remove “affected” after COVID-19;
- Line 23, please spell out the acronym of “ SPOKE”;
- Line 26, please change “COVID-19 infection” to “ COVID-19 symptoms”;
- Line 36, please change “ infectious” to “infected”;
- Line 38, please change “ one’s own” to “ healthcare professional”;
- Line 40, please change “affected by” to “with“;
- Line 41, please removed “ affected” after COVID-19;
- Line 58, please removed “to” after “allowing”;
- Line 74, please change “cum granos alis” to English;
- Line 89, I would add “ positive” in front of “ COVID-19”;
- Line 91, please remove “infection”. COVID-19 is a disease caused by SARS-CoV-2.
- Line 97, please change “provided” to “provide”;
- Line 139, please change “contagion” to “ infection”;
- Line 148, please change “contagion spreading” to “pandemic”;
- Line 225, I would change “ contagion-free” to “COVID-19 free”;
- Line 227-228, please rephase the sentence;
- Line 234, please change “ the contagion spreading” to “ the spread of viruses”;
- Line 292, please change “ for” to “of ”;
- Line 333, please change “ changed” to “change”;
- Line 335, please change “ was” to “ is”;
- Line 343, please change “ developed” to “develop” and change “infection” to “ symptoms”;
Reviewer 3 Report
This manuscript entitled "A guide for oncologic patient management during COVID-19 pandemic: the initial experience of an Italian oncologic HUB with exemplificative focus on uro-oncologic patients." by Francesco A. Mistretta et al reported the a guide for oncologic patients during COVID-19 pandemic. Overall, this manuscript is well-written and well-organized and provides significant guide for management of uro-oncologic patients.
I have some questions.
- HUB stands for high-experienced hospital and this abbreviation should appear at first time.
- What is SPOKE ?
- From 8th of March until 23rd of April, 22 patients candidate for surgery were diagnosed for SARS-CoV-2 infection after nasopharyngeal swab and their hospitalization was annulled. --> How do you manage such patients and what are the outcomes ?
- Why do you measure cutaneous temperature but no ear (tympanic) temperature or others as cutaneous temperature may be affected by various factors.